# Performance of indirect adherence measures for daily oral pre-exposure prophylaxis for HIV among adolescent men who have sex with men and transgender women in Brazil

**Diana Zeballos** [1] *, **Laio Magno**[1,2], **Fabiane Soares**[1], **Jony Arrais Pinto Junior**[3], **Leila Amorim**[1,4], **Dirceu Greco**[5], **Alexandre Grangeiro**[6], **Inês Dourado**[1], **on behalf of The PrEP15-19 study group**[¶]

1 Instituto de Saúde Coletiva, Universidade Federal da Bahia, Salvador, Bahia, Brazil, 2 Departamento de Ciências da Vida, Universidade do Estado da Bahia, Salvador, Bahia, Brazil, 3 Departamento de Estatística, Universidade Federal Fluminense, Niterói, Rio de Janeiro, Brazil, 4 Instituto de Matemática e Estatística, Universidade Federal da Bahia, Salvador, Bahia, Brazil, 5 Faculdade de Medicina, Universidade Federal de Minas Gerais, Belo Horizonte, Minas Gerais, Brazil, 6 Departamento de Medicina Preventiva da Faculdade de Medicina da Universidade de São Paulo, São Paulo, São Paulo, Brazil

¶ Membership of the PrEP15-19 study group is provided in the Acknowledgments.
* dianazeballos@gmail.com

**Data Availability Statement:** Zeballos Rivas, Diana Reyna, 2024, "Performance of indirect adherence

## Abstract

### Background

Consistent monitoring of PrEP adherence with accurate measurement tools at point-of-care could greatly contribute to reaching adolescents with poor adherence. We aimed to assess the performance of indirect adherence measures to oral PrEP among adolescent men who have sex with men (AMSM) and adolescent transgender women (ATGW).

### Methods

PrEP15-19 is a prospective, multicenter, PrEP demonstration cohort study that includes AMSM and ATGW aged 15–19 in three Brazilian cities. A diagnostic accuracy study was conducted using tenofovir-diphosphate (TFV-DP) concentrations in dried blood spots as the reference standard, along with three index tests: medication possession ratio (MPR), pill count, and self-report. We calculated the area under the curve (AUC) for protective TFV-DP levels ($\geq$800 fmol/punch) and sensitivity (SE) and specificity (SP) for established cutoff points.

### Results

We included 302 samples from 188 participants. Most of participants were AMSM (78.7%), aged 18–19 years (80.3%), and non-whites (72.9%). The AUC was 0.59 for MPR, 0.69 for pill count, and 0.75 for self-report. When combining MPR and self-report, the AUC increased to 0.77. Sensitivity was high for the cutoff points identified by the Youden index, 80% for

measures for daily oral pre-exposure prophylaxis
for HIV among adolescent men who have sex with
men and transgender women in Brazil.", https://doi.
org/10.7910/DVN/SOBKYM, Harvard Dataverse
The authors also confirm that they did not have any
special access or request privileges that others
would not have "The data relevant to this paper is
available from the Harvard Dataverse at https://doi.
org/10.7910/DVN/SOBKYM."

**Funding:** This project was funded by Unitaid
[2017-15-FIOTECPrEP] in the form of a grant to ID
and by the Brazilian National Council for Scientific
and Technological Development (CNPq) [141796/
2019-7] in the form of a doctoral fellowship to DZ.

**Competing interests:** The authors have declared
that no competing interests exist.

MPR, 92% for self-report, and 97% for pill count. However, specificities were low 40%, 46%, and 38%, respectively.

## Conclusions

Indirect measures were able to discriminate adolescents with good adherence. However, their performance in identifying those with low adherence might be limited, suggesting that it is necessary to initiate adherence interventions when there is no evidence of perfect adherence. Combining measures can provide wider information on adherence.

## Introduction

Oral pre-exposure prophylaxis for HIV (PrEP) with tenofovir disoproxil fumarate plus emtricitabine (FTC/TDF) is a safe and effective strategy to reduce new HIV infections among adolescents and young people [1, 2]. In 2019, PrEP was available in 77 countries, with about 626,000 people who received at least the first PrEP prescription, 69% more than in 2018 [3]. Brazilian national public health system (in Portuguese- Sistema Único de Saúde- SUS) incorporated oral PrEP in 2017 for populations at higher risk of HIV infection [4]. In 2022, adolescents aged 15 to 17 years old were included in the updated Brazilian PrEP guidelines, expanding the prevention options for this population [5].

Unlike antiretroviral therapy, PrEP has not a lifetime indication and is recommended during periods of high risk of infection, when adherence to medication is key for PrEP to work [6]. Daily oral PrEP reduces the risk of HIV infection by 96% when at least four pills per week are used [7], however achieving and sustaining protective levels of PrEP has proven to be challenging among adolescents and young individuals [1, 8, 9]. A systematic review revealed that the proportion of suboptimal adherence among youth from sexual and gender minorities was 57.0%, which is higher than the overall proportion of 41.0% and the proportion among older individuals, which was 29.1% [10]. Continuous adherence monitoring could significantly help to reach populations with lower adherence rates, thus accurate adherence measures at point of care to capture this are needed.

Different measures have been used to assess adherence to PrEP, but there is no stablished "gold standard". Each method has its strengths and limitations [11–13]. Direct measures involve quantifying the drug in blood, cells, hair, or urine and are deemed more accurate, which is why they are frequently employed as a reference standard [13]. That is the case of tenofovir-diphosphate (TDF-DP) quantification in dried blood spots (DBS) for PrEP studies. However, its high cost and implementation complexity limit its use in the routine of health services [14]. Indirect measures that evaluate adherence by asking the PrEP users or tracking pharmacy refills and drug dispensation records emerged as a point of care options for implementation during PrEP monitoring.

Self-report is widely used, easy to collect, and low-cost. However, the concern with this measure is the overestimation of adherence due to social desirability bias or recall bias [15]. Pill count is calculated based on the pills dispensed and returned. Medication possession ratio (MPR) is estimated from pharmacy records, considering the days between visits and pills dispensed. Both measures are easy to calculate and low-cost, the limitation with these measures is that we are assessing PrEP coverage and assuming that the pills were used [11]. Studies that compared indirect measures with DBS have found that MPR, pill count, and self-report can discriminate participants with and without sufficient drug levels for protection against HIV

infection [14, 16, 17], however, few studies assessed the value of these measures among adolescents. One study conducted with young MSM, reported that self-report had a low discrimination capacity [18].

Assessing adherence is fundamental to timely identify individuals who require additional support for PrEP adherence. Furthermore, there is a need for feasible and affordable methods to assess adherence among adolescents and youth. Our objective was to evaluate the performance of indirect measures for daily oral PrEP adherence, such as MPR, pill count, and self-report, compared to TFV-DP levels in DBS, among adolescent men who have sex with men (AMSM) and adolescent transgender women (ATGW) participants of the PrEP1519 study in Brazil.

## Methods

### Study design, setting, and participants

We conducted a diagnostic accuracy study using data from PrEP15-19, a PrEP demonstration cohort study of AMSM and ATGW aged 15–19 at high risk for HIV infection, residing in Salvador, Belo Horizonte or Sao Paulo, three large capital cities in Brazil. Participants self-selected into one of two arms: i) the PrEP arm, which included those who enrolled in daily oral PrEP with the TDF/FTC combination; ii) the non-PrEP arm, which included those who were eligible for PrEP but chose not to use drug prophylaxis. Both groups also received other HIV combination prevention methods (i.e., counseling, condoms, lubricant, and HIV self-test). To be included in the PrEP arm, participants must had a negative HIV test and meet at least one of the following criteria: (i) having had condomless anal sex in the past six months; (ii) having had one or more episodes of STIs in the last 12 months; (iii) having used PEP in the last 12 months; (iv) have used alcohol and/or other drugs during sexual intercourse; (v) having engaged in commercial sexual relations; (vi) having suffered violence and discrimination due to their sexual orientation. Participants received oral PrEP in a combined pill with a fixed dose of emtricitabine 200 mg (FTC) and tenofovir disoproxil fumarate 300 mg (TDF), to be taken once daily. Following visits were scheduled at baseline, 30 days, 60 days, and then every 90 days thereafter, until the end of the study in February 2022. At each visit, enough PrEP bottles containing 30 pills were dispensed to cover the days until the next visit. In addition, at each follow-up visit, a dried blood spot (DBS) sample was collected and stored to measure direct PrEP adherence afterward. More details of the study have been published elsewhere [19].

For this analysis, we included participants who initiated PrEP and had at least one follow-up visit where a DBS sample was collected and stored between February 21, 2019, in São Paulo, April 2, 2019, in Salvador, May 13, 2019, in Belo Horizonte, and December 18, 2020, at all three sites.

### Adherence measures

We used a direct measure as reference standard, the quantification of TFV-DP in DBS samples, while index tests were indirect measures including medication possession ratio (MPR), pill count, and self-report adherence.

**Medication possession ratio.** MPR was calculated using pharmacy refill records and defined as the ratio between the number of pills dispensed and the number of days between visits. MPR ranges from zero to 1. However, this ratio can exceed 1 if more medication was dispensed than needed for the period. Values equal or more than 1 indicate being full covered during the period ($\geq$100%).

**Pill count.** Participants were asked to return their medication bottles at each follow-up visit, and the number of unused pills was counted. Pill count was then calculated as the

difference between the number of pills dispensed and the number of pills returned, divided by the number of days between visits. Results were reported as percentages, with higher values indicating better adherence.

**Self-report adherence.** Self-report was assessed during clinical assessments using the following question: 'During the last month, approximately how many days have you missed your PrEP pills? Even if it has been one or many days, please tell me as it will not affect your care.' We then subtracted the number of missed days from 30 and calculated the percentage, with 100% indicating perfect adherence.

*TFV-DP concentrations in DBS.* During all follow-up visits, blood was collected and spotted onto filter paper for DBS, and then the DBS samples were stored. The quantification of TFV-DP concentrations in DBS was conducted at the University of Colorado Antiviral Pharmacology Laboratory (Aurora, CO, USA) using liquid chromatography mass spectrometry tandem mass spectrometry (LC-MS/MS) and extracted with 50% methanol solution. A TFV-DP concentration equal to or greater than 800 fmol/punch corresponded to 4 doses/week, as reported by laboratory. TFV-DP levels were dichotomized into highly protective drug levels (>800 fmol/punch) and poorly protective drug levels (<800 fmol/punch) [20, 21].

## Sample size calculation

The sample size was defined using Tilaki's [22] approach who proposed a calculation for validation studies in the field of health. Using a pre-established AUC of 0.7 and a marginal error of 0.10, the required sample size was 108 adolescents for each group (highly protective drug levels and poorly protective drug levels), with 80% statistical power and 95% confidence level. The quantification of TFV-DP concentrations was essayed in stored DBS samples from i) all the participants who seroconverted to HIV while receiving PrEP; ii) all ATGW, given the smaller sample size compared to AMSM; and iii) a random sample of DBS from AMSM. The DBS samples from AMSM were numbered chronologically according to the visit date, generating the follow-up visit number. We first listed the DBS samples by site and follow-up visit number, and then DBS samples were randomly selected.

## Statistical analysis

A descriptive analysis was conducted by subpopulation using the chi-square test to compare characteristics of participants included and not included. DBS data were matched with data from each indirect measure collected on the same date. If the same date was not available, we used the closest data within a 45-day range. Information of participants who did not return their PrEP bottles or did not answer the self-report question were excluded as missing data for the corresponding indirect measure and time. We compared drug levels between missing data for indirect measures and complete information using Generalized Estimating Equations (GEE) with the logit link function. The compound symmetry correlation structure was adopted for GEE. The discriminatory capacity of each indirect measure and of the combination of these measures was assessed through the computation of the Area Under the Curve (AUC) using the Receiver Operating Characteristics (ROC) curve. The optimal cutoff points were estimated using the Youden index, which identified the points with the best balance of sensitivity and specificity [23]. We compared two approaches for this analysis, the first approach was conducted using only the first available measure per participant. In the second approach we used the repeated measures for the same participant, modeling through GEE [24, 25]. As the cutoff points in the second approach are based on the probabilities of adherence estimated by the GEE adjusted from the indirect measure, we concluded that cut-off points would not be easy to apply in clinical practice. Therefore, we opted for the alternative

naive approach, which involved using a single measure to estimate the cutoff points. This decision was supported by the similarity of results for AUC in both approaches. Sensitivity (SE), specificity (SP), and positive (PPV) and negative (NPV) predictive values were calculated for the best cut-off points identified by Youden Index and for cutoff points equivalent to 4-day per week use (0.6 for MPR and 57.1% for pill count and self-report) and 7-day per week use ($\geq 1.0$ for MPR and 100% for pill count and self-report). STATA software version 15 (StataCorp, 2015) and R version 4.3.1 were used for the analyses.

### Ethical issues

The PrEP1519 study was conducted according to the Brazilian (Resolution CNS no. 466, Brazil, 2012) and international research ethics guidelines, and was approved by the Research Ethics Committees (REC) of the World Health Organization (Protocol ID: Fiotec-PrEP Adolescent study), Federal University of Bahia (#3,224,384), University of São Paulo (#3,082,360) and Federal University of Minas Gerais (#3,303,594). Written informed consent (WIC) was obtained from adolescents 18–19 years. For those <18 years, each site had a different protocol according to local court decisions: in Belo Horizonte the WIC had to be signed by the parents or guardians followed by the assent form (AF) signed by the adolescents; in Salvador there were two options: i) WIC signed by a parent or guardian and AF by the adolescent; or ii) only AF signed by the adolescent when the sociopsychology team judged that their family ties had been severed or that they were at risk of physical, psychological, or moral violence due to their sexual orientation; and in São Paulo only the AF signed by the adolescents was required. All participants could withdraw consent at any stage of the process or skip any questions perceived as too sensitive, too personal, or distressing.

### Results

A total of 703 participants have initiated PrEP during the study period, 1,447 specimens of DBS were collected from AMSM and 89 from ATGW. Out of these, 302 (19.6%) DBS samples were sent for the quantification of TDF-DP, 32 samples from individuals who seroconverted, 86 samples from ATGW, and 184 samples from AMSM. The distribution of AMSM DBS samples by follow-up visit number and week of follow-up are displayed in S1 Table and the distribution of DBS Samples by collection week for all the samples are displayed in S2 Table. These DBS samples were obtained from a total of 188 adolescents, which corresponded to 26.7% of all participants enrolled in PrEP. Most of the participants included in this analysis were AMSM (78.7%), non-white (72.9%), and aged 18–19 years (80.3%) (Table 1). When comparing the characteristics of those who were included versus those who were not, significant differences were found in the study site for both AMSM (p = 0.019) and ATGW (p = 0.004) subpopulations (S3 Table). Since all DBS samples from ATGW were sent for quantification, the "not included" category for TGW refers to those who either started PrEP but did not attend follow-up visits or had DBS samples that were lost.

Matched data with indirect measures were 302 for MPR, 274 for self-report, and 104 for pill count. We observed poorly protective drug levels for majority of visits where the PrEP bottle was not returned (67.68%), and when the self-reported adherence question was not answered (82.14%) (Table 2).

The analysis of the ROC curve indicates that the three measures were able to discriminate those with highly protective levels of TDF-DP with AUC of 0.59 for MPR, 0.69 for pill count, and 0.75 for self-report (Fig 1 and Table 3).

Additionally, we observed that combining MPR with self-report resulted in a better discrimination capacity (AUC = 0.77). When we performed the analysis using the first measure

**Table 1. Baseline characteristics of participants included in the accuracy analysis.** PrEP1519 study, February 2019 to December 2020.

| Characteristics | Total |
|---|---|
| Age | |
| 15–17 years | 37 (19.68) |
| 18–19 years | 151 (80.32) |
| Sub-population | |
| Men who have sex with men | 148 (78.72) |
| Transgender women | 40 (21.28) |
| Skin-color | |
| White | 51 (27.13) |
| Non-White | 137 (72.87) |
| Study site | |
| Belo Horizonte | 53 (28.19) |
| Salvador | 64 (34.04) |
| São Paulo | 71 (37.77) |
| Schooling | |
| Higher education | 44 (23.40) |
| High school or less | 142 (75.53) |
| Not Available | 2 (1.06) |
| Condomless anal sex | |
| No | 63 (33.51) |
| Yes | 125 (66.49) |
| Partner living with HIV | |
| No/Don't know | 148 (78.72) |
| Yes | 13 (6.91) |
| Not Available | 27 (14.36) |

of adherence of each participant, we found results similar to those found with repeated measures (Table 3).

The best cut-off points identify by Youden Index were 0.91 for MPR, 83.30% for self-report, and 58.70% for a pill count. Sensitivity, specificity, and predictive values for each cut-off point are displayed in Table 4.

As predictive values are influenced by prevalence, we also estimated the predictive values for different scenarios of adherence prevalence (Table 5).

**Table 2. Comparison of included and not included values due to missing data by levels of TFV-DP in DBS.** PrEP1519 study, February 2019 to December 2020.

| Characteristics | Total | Quantification of TFV-DP in DBS | | |
|---|---|---|---|---|
| | | >4 days/week | < 4 days/week | p value |
| Self-report | | | | 0.016 |
| Included | 274 (90.73) | 110 (40.15) | 164 (59.85) | |
| Not Included | 28 (9.27) | 5 (17.86) | 23 (82.14) | |
| Pill-count | | | | 0.042 |
| Included | 104 (34.44) | 51 (49.04) | 53 (50.96) | |
| Not Included | 198 (65.56) | 64 (32.32) | 134 (67.68) | |

TFV-DP = tenofovir-diphosphate. DBS = dried blood spot.

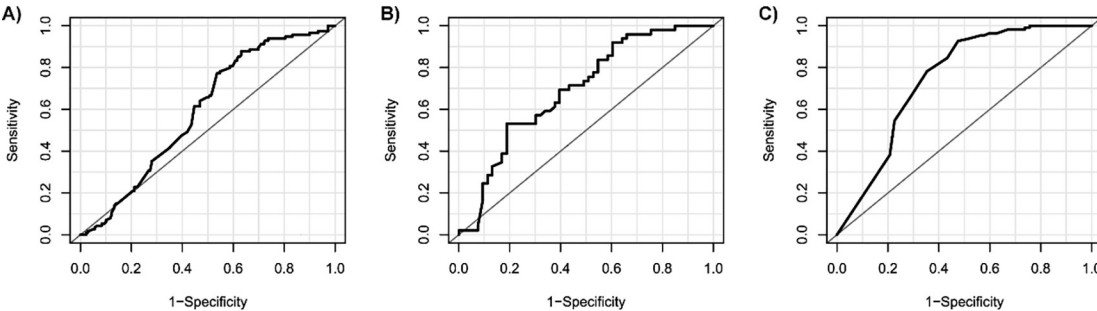

**Fig 1.** Receiver Operating Characteristics curves for indirect measures of adherence A) Medication Possession Ratio (n = 188), B) pill count (n = 68) and C) Self-report (n = 174) considering first DBS per participant. PrEP1519 study, February 2019 to December 2020.

## Discussion

Indirect measures for monitoring PrEP adherence among adolescents have proven effective in identifying individuals who maintained sufficient PrEP levels to prevent HIV infection at various follow-up points. Among these measures, self-report demonstrated the highest performance. Combining different adherence measures, such as self-report and MPR, provides a more comprehensive understanding of an adolescent's adherence to PrEP. In selecting a cut-off point to determine adequate adherence using indirect measures, we found that a threshold equivalent to taking four pills per week offered high sensitivity but low specificity. This suggests that while indirect measures are effective in identifying those with good adherence, they are less capable of detecting those with poor adherence. To accurately identify adolescents with poor adherence, a cut-off point equivalent to perfect adherence would be necessary. Given their low cost and proven ability to differentiate users with high protective levels, these indirect measures can be valuable as point-of-care screening tools in PrEP programs, though it is essential to consider their limitations.

Although the literature on PrEP adherence often describes self-report as a method with poor performance that overestimates adherence, we found that self-report was the most effective measure in identifying adolescents who maintained enough PrEP levels to prevent HIV infection. In the iPrEX study among MSM and TGW, two self-report adherence collection strategies were employed, through yielding AUCs of 0.51 and 0.52 [26]. However, in the PrEP Brazil study, self-reports were able to discriminate participants with high protective levels

**Table 3. Accuracy of indirect adherence measures by itself and in combination, according to the first and repeated measures.** PrEP1519 study, February 2019 to December 2020.

| Adherence measures | n | First measure | n | Repeated measures |
|---|---|---|---|---|
|  |  | AUC (95%CI) |  | AUC (95%CI) |
| MPR | 188 | 0.55 (0.47–0.64) | 302 | 0.59 (0.53–0.66) |
| Pill count | 68 | 0.69 (0.57–0.82) | 104 | 0.69 (0.58–0.80) |
| Self-report | 174 | 0.72 (0.65–0.80) | 274 | 0.75 (0.68–0.81) |
| MPR + Pill Count + Self-report | 66 | 0.72 (0.59–0.84) | 102 | 0.72 (0.62–0.83) |
| Pill Count + Self-report | 66 | 0.71 (0.58–0.84) | 102 | 0.73 (0.63–0.83) |
| MPR + Pill Count | 68 | 0.68 (0.55–0.81) | 104 | 0.69 (0.58–0.80) |
| MPR + Self-report | 174 | 0.74 (0.67–0.82) | 274 | 0.77 (0.70–0.83) |

AUC = area under the curve; CI = Confidence interval; MPR = medication possession ratio.

**Table 4. Performance of cutoff points for indirect measures of PrEP adherence.** PrEP1519 study, February 2019 to December 2020.

| Adherence measures | Cutoff point | Sensitivity | Specificity | PPV | NPV |
|---|---|---|---|---|---|
| **Youden index** | | | | | |
| MPR | 0.91 | 0.80 | 0.40 | 0.48 | 0.74 |
| Pill count | 58.70 | 0.97 | 0.38 | 0.61 | 0.93 |
| Self-report | 83.30 | 0.92 | 0.46 | 0.55 | 0.89 |
| **Equivalent to 4-days-week use or posse** | | | | | |
| MPR | 0.60 | 0.95 | 0.17 | 0.44 | 0.83 |
| Pill count | 57.10 | 0.97 | 0.38 | 0.61 | 0.93 |
| Self-report | 57.10 | 0.96 | 0.30 | 0.50 | 0.91 |
| **Equivalent to 7-days-week use or posse** | | | | | |
| MPR | 1.00 | 0.47 | 0.57 | 0.43 | 0.62 |
| Pill count | 100.00 | 0.20 | 0.91 | 0.70 | 0.53 |
| Self-report | 100.00 | 0.37 | 0.80 | 0.57 | 0.64 |

PPV = positive predictive value. NPV = negative predictive value. MPR = medication possession ratio.

(AUC = 0.65) [14]. The findings of a systematic review revealed no important difference between suboptimal adherence as measured by self-reports and tenofovir concentrations, suggesting that self-reports act as a convenient and affordable approximation to PrEP adherence [10]. We hypothesized that the PrEP1519 multidisciplinary team created a trustworthy friendly environment where participants felt confident in disclosing their adherence without fear of being judged. This environment was further emphasized by peer navigators who helped improve linkage to services [27]. Self-report is a simple and inexpensive method, making it feasible to be used to assess adherence. Additionally, the quality of information obtained through self-report measures could be enhanced using strategies to reduce desirability bias or facilitate recall [15].

Studies that evaluated the performance of MPR used data from specific time periods, for example, PrEP Brazil assessed participants who attended the 48th week [14], and iPrEx used data from the 24-week [26], both studies showed that MPR has a better ability to discriminate drug detection levels than self-report and pill count. In contrast, we used data from different time points, which could have impacted its performance, due to the periods of discontinuation and restarting of the PrEP use.

One disadvantage of using pill-count is compliance with returning the bottles. This behavior may prevent providers from calculating adherence [13]. We observed that participants who

**Table 5. Predictive values according with variance in prevalence of PrEP adherence.**

| Adherence Prevalence (%) | MPR | | Pill count | | Self-report | |
|---|---|---|---|---|---|---|
| | PPV | NPV | PPV | NPV | PPV | NPV |
| 10 | 0.06 | 0.88 | 0.02 | 0.89 | 0.02 | 0.85 |
| 20 | 0.14 | 0.76 | 0.05 | 0.78 | 0.04 | 0.72 |
| 25 | 0.17 | 0.70 | 0.07 | 0.73 | 0.06 | 0.65 |
| 30 | 0.21 | 0.65 | 0.09 | 0.68 | 0.07 | 0.59 |
| 40 | 0.29 | 0.54 | 0.13 | 0.58 | 0.11 | 0.49 |
| 60 | 0.48 | 0.34 | 0.25 | 0.38 | 0.22 | 0.30 |
| 80 | 0.71 | 0.16 | 0.48 | 0.18 | 0.43 | 0.14 |

PPV = positive predictive value. NPV = negative predictive value. MPR = medication possession ratio.

did not return their bottles were more likely to have drug levels below the protective threshold. In the iPrEx study, pill count performed poorly [26], although in PrEP Brazil it was able to discriminate participants with good adherence [14]. A study using pill count with unannounced visits to participants also found that it performed poorly (AUC = 0.54) and had a poor correlation with more objective measures [28].

We observed an improvement in performance with the combination of MPR and self-report. Previous studies have evaluated the additional value of combining multiple adherence measures. For example, combining self-report with more accurate measures, such as quantification of TDF in hair or plasma, which may improve the ability to identify patients with good adherence [29]. Therefore, when possible, the use of multiple adherence measures is recommended, as different measures may increase the period of adherence being evaluated.

Sensitivity for cutoff points identified by Youden index were high (SE> = 0.8), indicating that most adolescents with high protective drug levels were identified as having high adherence by indirect measures. However, those cutoff points were less specific (SP<0.5), meaning that adolescents with poorly protective drug levels had a low probability of being identified as having poor adherence with indirect measures. The positive predictive values indicated that participants identified as having high adherence had probabilities of 48%, 55%, and 61% of having high drug protective levels, assessed with MPR, self-report, and pill count, respectively. Finally, indirect measures had high negative predictive values (NPV>0.7), meaning that when adolescents were identified as having poor adherence with indirect measures, the probability of having low protective drug levels was high. Furthermore, results showed that when the prevalence of adherence increases, the ability of indirect measures to identify adolescents with poor adherence will diminish.

Our study has some limitations. First, measures were paired with those collected on different days, although we tried to avoid large gaps between measurement times. Second, the limited number of repeated measures due to our sampling process prevented us from exploring longitudinal variables more extensively, besides there is a methodological challenge related to the interpretation of estimates obtained using GEE for clinical practice. Further statistical research and developments are needed to assess accuracy and calculate cut-off points when using repeated measures. Finally, the self-report question considered the last 30 days, even if the interval between visits was longer. Therefore, our results are specific to this timeframe. Different questions, such as those considering the entire period between visits or the last 7 days, will require their own validation.

## Conclusions

Adherence monitoring is critical to improve the effectiveness of PrEP. We found that self-report, MPR and pill count perform well in identifying adolescents taking enough PrEP to achieve adequate levels of protection. However, their performance in identifying those with low adherence might be limited, suggesting that it is necessary to initiate adherence interventions when there is no evidence of perfect adherence. Further research is needed to develop or identify measures that can detect adolescents with poor adherence or those at risk of poor adherence. In the meantime, self-report and MPR remain valuable tools for monitoring real-world PrEP use and for identifying adolescents who need additional support with PrEP adherence. Furthermore, we observed that a combination of measures will add value to the use of these measures, which are easy to implement at point-of-care PrEP programs within national health systems, such as in the Brazilian SUS.

## Supporting information

**S1 Table. Distribution of DBS collection weeks by follow-up visit number for randomly selected AMSM.**
(DOCX)

**S2 Table. Distribution of samples per weeks among all the samples of DBS.** PrEP1519 study, February 2019 to December 2020. [a]Interval notation is used to describe categories: parentheses indicate that the number is excluded from the interval, while square brackets indicate that the number is included in the interval.
(DOCX)

**S3 Table. Comparison of participants' baseline characteristics included and not included in the accuracy analysis, by subpopulation.** PrEP1519 study, February 2019 to December 2020. [a]Chi-square test. [b]Fisher test. [c]Not considered for the estimation of the association.
(DOCX)

## Acknowledgments

The authors would like to express their gratitude to the study participants, the local teams that carried out the fieldwork in the three cities, and to all the collaborators. The PrEP15-19 study group: Inês Dourado (ines.dourado@gmail.com), Dirceu Greco, Alexandre Grangeiro, Laio Magno, Fabiane Soares, Priscilla Caires, Joilson Paim, Lorenza Dezanet, Diana Zeballos, Unaí Tupinambás, Mateus Westin, Paula Massa, Beo Oliveira Leite, Gustavo Costa, Maria Mercedes Escuder.

## Author Contributions

**Conceptualization:** Diana Zeballos, Laio Magno, Alexandre Grangeiro, Inês Dourado.

**Data curation:** Diana Zeballos, Fabiane Soares, Jony Arrais Pinto Junior, Leila Amorim.

**Formal analysis:** Diana Zeballos, Jony Arrais Pinto Junior, Leila Amorim.

**Funding acquisition:** Dirceu Greco, Alexandre Grangeiro, Inês Dourado.

**Investigation:** Laio Magno, Fabiane Soares.

**Project administration:** Laio Magno, Fabiane Soares, Dirceu Greco, Alexandre Grangeiro, Inês Dourado.

**Software:** Jony Arrais Pinto Junior.

**Supervision:** Laio Magno, Dirceu Greco, Alexandre Grangeiro, Inês Dourado.

**Validation:** Jony Arrais Pinto Junior, Leila Amorim.

**Visualization:** Diana Zeballos.

**Writing – original draft:** Diana Zeballos.

**Writing – review & editing:** Diana Zeballos, Laio Magno, Fabiane Soares, Jony Arrais Pinto Junior, Leila Amorim, Dirceu Greco, Alexandre Grangeiro, Inês Dourado.

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
