## [Decision Letter · Decision Letter 0]

2 Jul 2024

PONE-D-24-10207Performance of indirect adherence measures for daily oral pre-exposure prophylaxis for HIV among adolescent men who have sex with men and transgender women in Brazil.PLOS ONE

Dear Dr. Zeballos,

Thank you for submitting your manuscript to PLOS ONE. After careful consideration, we feel that it has merit but does not fully meet PLOS ONE’s publication criteria as it currently stands. Therefore, we invite you to submit a revised version of the manuscript that addresses the points raised during the review process.

We look forward to receiving your revised manuscript.

Kind regards,

Fengyi Jin, Ph.D.

Academic Editor

PLOS ONE

Journal Requirements:

"This project was made possible by funding and support from UNITAID (#2017-15-FIOTECPrEP)"

4. One of the noted authors is a group or consortium "The PrEP1519 study group". In addition to naming the author group, please list the individual authors and affiliations within this group in the acknowledgments section of your manuscript. Please also indicate clearly a lead author for this group along with a contact email address.

Reviewers' comments:

Reviewer's Responses to Questions

**Comments to the Author**

1. Is the manuscript technically sound, and do the data support the conclusions?

Reviewer #1: Partly

Reviewer #2: Yes

Reviewer #3: Partly

2. Has the statistical analysis been performed appropriately and rigorously? 

Reviewer #1: Yes

Reviewer #2: I Don't Know

Reviewer #3: No

3. Have the authors made all data underlying the findings in their manuscript fully available?

Reviewer #1: Yes

Reviewer #2: No

Reviewer #3: Yes

4. Is the manuscript presented in an intelligible fashion and written in standard English?

Reviewer #1: No

Reviewer #2: Yes

Reviewer #3: Yes

5. Review Comments to the Author

Reviewer #1: The authors aim to evaluate the performance of MPR, pill count, and self-report, compared to referent DBS, among adolescent men have sex with men and adolescent transgender women in Brazil. The study makes important contributions by examining how alternative metrics to assess PrEP adherence could be used in resource limited settings. Ultimately, though, much more detail about the study design and analytic strategy is needed to better interpret the results and contextualize these findings.

Major suggested revisions:

1. Authors report prior literature in the introduction that “Studies that compared indirect measures with DBS have found that MPR, pill count, and self-report can discriminate participants with and without sufficient drug levels for protection against HIV infection [12, 14, 15]” which is the goal of this study – more information is needed on the gap in knowledge to assess what this study is adding to the literature and why it is important.

2. MPR seems like a valuable tool for PrEP persistence rather than PrEP adherence – I’m unsure how someone’s maximal PrEP adherence (the function of MPR) is related to pill taking and ultimately adherence. The use of this metric needs more justification in the context of pill taking and adherence in this study.

3. The sampling mechanism for DBS is quite confusing and is presented in a way that I (personally) cannot follow. Did all participants have DBS measured all time points? What is the justification for 30 samples for each visit number? Could participants who had a sample taken at the first visit not have a sample taken at the second visit, and then a sample at the third visit? These questions would further impact how you model these data for your longitudinal analyses and impact your interpretation of results. Further, I would present the power analysis before discussing the sampling that took place.

4. There is no mention of controlling for confounding factors in this analysis. Were these accounted for? There is differential sociodemographic characteristics in both groups (aMSM aTGW) which would impact self-report measures. Further, the variability in the informed consent process would significantly introduce selection bias as participants in some regions did not have to have parental consent and should be accounted for. Were these metrics controlled for?

5. Table 1 could be modified to remove values for persons not included in the study and moved to a supplemental text. Why were these specific participants not included? A breakdown either by figure or table for numbers of persons excluded by inclusion/exclusion criteria would be helpful to understand why individuals were not selected to participate in the study. Further, does the “not included” participants column(s) include participants that were not in the PrEP arm? Further, I do not think this table needs to be stratified by gender identity as you do not provide estimates stratified by gender identity further in the text.

6. It seems as those these metrics are great for identifying individuals who are adherent to PrEP but not so much those who are non-adherent. If this is a way to identify persons who are non-adherent, there needs to be more integration of existing literature on poor-adherence metrics or better ways to capture this specific population.

Minor suggested revisions

7. It is unclear what authors mean by “first dose” [line 29]. Is this literally the first pill, or their first prescription? Are these first time PrEP users?

8. Participants self-selected into arms which may introduce a selection bias as participants who chose to participate in the study and chose to take PrEP may be more inclined to use daily PrEP properly and should be noted as a limitation in the discussion.

9. It is unclear if inclusion/exclusion criteria [line 77-82] are for the larger study data or are specific to these set of analyses. If they are for all participants of the larger study (not just PrEP users) I would consider moving them above the citation for the parent study.

10. Follow up visits were scheduled at baseline, 30 days, 60 days, and quarterly thereafter [line 84-85] but there is no mention of how many visits occurred past the 60 day mark. Please define quarterly in number of days as the prior to visits were also described as such and include a specific number of visits for the study duration.

11. It is stated that MPR values range from 0-1; however, it is also stated that values equal or more than 1 indicate that values could range beyond 1? Please clarify either the range of MPR values, or the set of values that indicate full coverage.

12. Pill count seems like a false representation of the actual metric being assessed – this is a proportion of pills used per all pills received during a given time period, not a simple count. An alternative title for this metric may help the reader to better understand the true use of this metric.

13. It may be beneficial to add another level of subheadings for each of your metrics for assessing PrEP use.

14. In your sample size calculation it would be beneficial to get the exact numbers of those who seroconverted, number of TGW, and how many MSM were sampled at this stage as a direct comparison for the calculated power (in this section). The proportion of samples in each of these three groups could impart bias even if a random sample of aMSM was conducted.

15. There is no information on model type that was used to estimate ROC. GEE is an estimation method used in tandem with a specific model (linear, logistic). Was logistic regression used to build the ROC curves with GEE for longitudinal data analyses? Further, please specify which correlation structure was used with GEE (which should align with sampling methods and a priori hypothesis) as this can impact precision.

16. I would add a citation for the Youdon index and a brief explanation of the purpose.

17. Throughout, sensitivity is sometimes referred to as sensibility and sometimes referred to as sensitivity. Please adjust all mentions of sensibility to sensitivity.

18. You can not have over 100% for the pill count and self report as it is currently described. The “≥” [Line 150] should be changed to >.

19. Please clarify what NA stands for in Table 1. Are these missing?

20. Please provide 95% confidence intervals for all estimates of AUC throughout the text considering the sample size is relatively small.

21. I would suggest supplemental tables that include values for the GEE analyses for Tables 4 and 5 to assess differences in the analytical methods. It seems as though overall in longitudinal analyses there are larger AUCs which may be reflective of behavior over time and if persons are being seen by the same provider, these metrics can be assessed.

22. There exist instances in the text where aTGW and aMSM are used and others where ATGW and AMSM are used. Please use one acronym as to not confuse readers.

23. Please

Reviewer #2: This is a useful study and adds to the body of knowledge about what is known about PrEP effectiveness and measuring adherence. I suggest the following minor revisions:

1. The abstract should contain the results in digestible format that is easily understood; this can be done by reporting in percentages and stating clearly in the conclusion which indirect measures are most useful/recommended based on the results from the AUCs.

2. Authors might need to check the acronyms and select between between referring to adolescent MSM and TGW as aMSM or AMSM (aTGW or ATWG) to avoid confusion.

3. I suggest that the conclusion be strengthened along the lines suggested in point 1. This can be done by clearly identifying which indirect measures are most useful/easy to implement and add as recommendations.

Great job!

Reviewer #3: This study utilized tenofovir-diphosphate (TFV-DP) concentrations as the reference standard and evaluated three index tests: medication possession ratio (MPR), pill count, and self-report. The area under the curve (AUC) was calculated for protective TFV-DP levels (≥800 fmol/punch), while sensitivity (SE) and specificity (SP) were employed for established cutoff points.

Areas for potential improvement in this study include:

The term "unprotected sex" in line 78 is not specific; it is suggested to modify it to specialized terms like "condomless anal intercourse."

Self-report was assessed based on a question about missing doses during the last month, but the follow-up intervals were not consistently 30 days. It is recommended to add relevant explanations in the discussion of limitations.

MPR ranges from zero to 1, with values equal to or greater than 1 indicating full coverage during the period (≥100%). The range of MPR was stated as 0-1 initially, so it is contradictory to describe values exceeding 1 later.

Consistency is needed in descriptions such as AMSM and aMSM, ATGW and aTGW.

The decimal places in Table 4 are inconsistent and should be revised for uniformity.

The calculation method for the P-values in Table 1 was not explained. It is inferred from the statistical methods section that the chi-square test was used, but due to low frequencies in variables like “schooling” and “partner living with HIV”, the Fisher's exact test would be more appropriate. Additionally, upon reevaluation, discrepancies were found in the P-values for “schooling” and “partner living with HIV” in the TGW subgroup compared to the authors' results.

It is recommended to merge the three curves in Fig 1 into one coordinate system, differentiate them with different colors, and clearly label the AUC values for each method.

It is suggested to include an introduction in the background section regarding the adherence levels of global adolescents using PrEP.

6. PLOS authors have the option to publish the peer review history of their article (what does this mean?). If published, this will include your full peer review and any attached files.

Reviewer #1: No

Reviewer #2: **Yes: **Dr. Helen Anyasi

Reviewer #3: No

---

## [Author Response · Author response to Decision Letter 0]

16 Aug 2024

August 16, 2024

Editorial Board

Plos One

Ref.: Manuscript ID PONE-D-24-10207 entitled “Performance of indirect adherence measures for daily oral pre-exposure prophylaxis for HIV among adolescent men who have sex with men and transgender women in Brazil.”

Dear Editorial Board Members:

We would like to thank you for the opportunity to resubmit a revised copy of our manuscript. We would also like to take this opportunity to express our thanks to the reviewers for the feedback and helpful comments for correction or modification. 

Please found below answer to the reviewers' comments:

Reviewer #1

The authors aim to evaluate the performance of MPR, pill count, and self-report, compared to referent DBS, among adolescent men have sex with men and adolescent transgender women in Brazil. The study makes important contributions by examining how alternative metrics to assess PrEP adherence could be used in resource limited settings. Ultimately, though, much more detail about the study design and analytic strategy is needed to better interpret the results and contextualize these findings.

Major suggested revisions:

1. Authors report prior literature in the introduction that “Studies that compared indirect measures with DBS have found that MPR, pill count, and self-report can discriminate participants with and without sufficient drug levels for protection against HIV infection [12, 14, 15]” which is the goal of this study – more information is needed on the gap in knowledge to assess what this study is adding to the literature and why it is important.

Answer: Thank you for highlighting that. We have added a phrase to emphasize the lack of studies among adolescents from key populations, as follows

“Studies that compared indirect measures with DBS have found that MPR, pill count, and self-report can discriminate participants with and without sufficient drug levels for protection against HIV infection [12,14,15], however, few studies assessed the value of these measures among adolescents.”

2. MPR seems like a valuable tool for PrEP persistence rather than PrEP adherence – I’m unsure how someone’s maximal PrEP adherence (the function of MPR) is related to pill taking and ultimately adherence. The use of this metric needs more justification in the context of pill taking and adherence in this study.

Answer: We consider MPR a proxy for adherence. As stated in the introduction, indirect measures have limitations but are easier to use. The limitation of MPR is that it evaluates drug possession rather than the actual act of taking the pill, thereby we are estimating days covered by PrEP. Simple measures are needed for use in clinics during follow-up, and the aim of this study was to assess whether MPR could help evaluate adherence among adolescents, as other studies have shown MPR's discriminatory capacity in other populations. We add more information about MPR in the introduction to clear this issue, as follows:

“Pill count is calculated based on the pills dispensed and returned. Medication possession ratio (MPR) is estimated from pharmacy records, considering the days between visits and pills dispensed. Both measures are easy to calculate and low-cost, the limitation with these measures is that we are assessing PrEP coverage and assuming that the pills were used.”

3. The sampling mechanism for DBS is quite confusing and is presented in a way that I (personally) cannot follow. Did all participants have DBS measured all time points? What is the justification for 30 samples for each visit number? Could participants who had a sample taken at the first visit not have a sample taken at the second visit, and then a sample at the third visit? These questions would further impact how you model these data for your longitudinal analyses and impact your interpretation of results. Further, I would present the power analysis before discussing the sampling that took place.

Answer: Thank you for your input. We have enhanced the description of the DBS collection and sampling. All participants had DBS measured at all time points. However, the cost of data quantification restricted us from performing it on all the DBS samples. The sample size of 30 for each visit was chosen as the minimum number needed to ensure a reliable estimate of sensitivity and specificity, according with the different scenarios calculated, however not all the follow-up visits had the same number of observations. We also add S1 and S2 Tables to show the distribution of sample selected per week. To gather information from all visits, we sampled DBS instead of individuals, which resulted in fewer repeated measures per participant. This limitation is addressed in the limitations section.

We followed revisor’s suggestion of placing sample calculation before sampling description, and we added the following statement.

“During all follow-up visits, blood was collected and spotted onto filter paper for DBS, and then the DBS samples were stored.”

4. There is no mention of controlling for confounding factors in this analysis. Were these accounted for? There is differential sociodemographic characteristics in both groups (aMSM aTGW) which would impact self-report measures. Further, the variability in the informed consent process would significantly introduce selection bias as participants in some regions did not have to have parental consent and should be accounted for. Were these metrics controlled for?

Answer: Thank you for your question. As we conducted a diagnostic accuracy study, our primary objective was to assess the performance of indirect measures of adherence. We acknowledge that indirect measures, such as pill counts and self-reports, have limitations and can be influenced by social desirability bias, which may vary with different sociodemographic characteristics. However, a detailed analysis of these factors is beyond the scope of this study. Regarding the variability in the informed consent process, it could introduce selection bias in the decision to start PrEP. However, our population includes adolescents who already start PrEP, so we believe this does not affect our analysis.

5. Table 1 could be modified to remove values for persons not included in the study and moved to a supplemental text. Why were these specific participants not included? A breakdown either by figure or table for numbers of persons excluded by inclusion/exclusion criteria would be helpful to understand why individuals were not selected to participate in the study. Further, does the “not included” participants column(s) include participants that were not in the PrEP arm? Further, I do not think this table needs to be stratified by gender identity as you do not provide estimates stratified by gender identity further in the text.

Answer: Thank you for the suggestion. In Table 1, we now present only the characteristics of the population included in the study. We moved the stratified table to the supplementary material (S3 Table). Actually, the "not included" column differs for MSM and TGW. For MSM, it refers to those who were not sampled for DBS. Since all DBS samples from transgender women were sent for quantification, the "not included" category for TGW refers to those who started PrEP but did not have follow-up visits or had lost DBS samples. We also add a text in the results section to explain it:

“Since all DBS samples from ATGW were sent for quantification, the "not included" category for TGW refers to those who either started PrEP but did not attend follow-up visits or had DBS samples that were lost.”

6. It seems as those these metrics are great for identifying individuals who are adherent to PrEP but not so much those who are non-adherent. If this is a way to identify persons who are non-adherent, there needs to be more integration of existing literature on poor-adherence metrics or better ways to capture this specific population.

Answer: Thank you for pointing that out. In fact, the concern you mention is a finding of our study. To address the limitations of indirect adherence measures identified in our results, we suggest initiating adherence interventions when perfect adherence is not evident and combining multiple measures to obtain a more comprehensive assessment during follow-up evaluations. Due to the nature of our outcome, we analyzed the data with non-adherence as the focus. However, results using ROC curves were consistent, and the term "non-adherence" was found to be confusing when presenting our results. Therefore, we decided to maintain the focus on adherence. We used specificity to explore the usefulness of identifying non-adherence and arrived at the initial suggestions. We do not have records of specific measures for non-adherence unless the self-report question, for example, is different.

Minor suggested revisions

7. It is unclear what authors mean by “first dose” [line 29]. Is this literally the first pill, or their first prescription? Are these first time PrEP users?

Answer: Thanks for this observation, we were referring to the first PrEP prescription. We corrected it in the text.

8. Participants self-selected into arms which may introduce a selection bias as participants who chose to participate in the study and chose to take PrEP may be more inclined to use daily PrEP properly and should be noted as a limitation in the discussion.

Answer: Thank you for your comment. We conducted a diagnostic accuracy study, including a random sample of DBS from MSM and all DBS from TGW and seroconversions, which we believe eliminates selection bias, for this kind of study. Besides, we observed that 62% of the included sample did not have protective levels of TFV, indicating low adherence, so we did not only have participants with high adherence. Our inclusion criterion was the use of PrEP, mitigating the concern that participants choosing to use PrEP or not could introduce selection bias. This potential bias would be more relevant if our study had a different research question, for example, related with factors influencing adherence.

9. It is unclear if inclusion/exclusion criteria [line 77-82] are for the larger study data or are specific to these set of analyses. If they are for all participants of the larger study (not just PrEP users) I would consider moving them above the citation for the parent study.

Answer: Thank you for your comment. We moved all the information of the larger study before the citation.

10. Follow up visits were scheduled at baseline, 30 days, 60 days, and quarterly thereafter [line 84-85] but there is no mention of how many visits occurred past the 60-day mark. Please define quarterly in number of days as the prior to visits were also described as such and include a specific number of visits for the study duration.

Answer: Thanks for this suggestion, we change it as follows: “Following visits were scheduled at baseline, 30 days, 60 days, and then every 90 days thereafter, until the end of the study in February 2022.” We don’t have a determined number of visits as it was an open cohort. 

11. It is stated that MPR values range from 0-1; however, it is also stated that values equal or more than 1 indicate that values could range beyond 1? Please clarify either the range of MPR values, or the set of values that indicate full coverage.

Answer: Thank you for your question. When the number of pills an adolescent had exceeded the number of days between visits, it could generate values above 1. This occurred frequently in our study. We clarify this in the text.

“MPR was calculated using pharmacy refill records and defined as the ratio between the number of pills dispensed and the number of days between visits. MPR ranges from zero to 1. However, this ratio can exceed 1 if more medication was dispensed than needed for the period. Values equal or more than 1 indicating indicate being full covered during the period (≥100%).”

12. Pill count seems like a false representation of the actual metric being assessed – this is a proportion of pills used per all pills received during a given time period, not a simple count. An alternative title for this metric may help the reader to better understand the true use of this metric.

Answer: Thank you for your comment. "Pill count" refers to the process of counting the pills in the bottle that the participant returned at each visit. We chose to retain this term due to its common use in adherence literature. We add an explanation in the introduction.

“Pill count is calculated based on the pills dispensed and returned.”

13. It may be beneficial to add another level of subheadings for each of your metrics for assessing PrEP use.

Answer: Thanks for this suggestion. We included subheadings.

14. In your sample size calculation, it would be beneficial to get the exact numbers of those who seroconverted, number of TGW, and how many MSM were sampled at this stage as a direct comparison for the calculated power (in this section). The proportion of samples in each of these three groups could impart bias even if a random sample of aMSM was conducted.

Answer: Thank you for the suggestion. Due to the small sample size, we were unable to perform stratified analyses by subpopulations. However, exploratory analyses indicated similar results. The number of samples for each group is detailed in the first paragraph of the results section, as follows: “Out of these, 302 (19.6%) DBS samples were sent for the quantification of TDF-DP, 32 samples from individuals who seroconverted, 86 samples from ATGW, and 186 samples from AMSM.”

15. There is no information on model type that was used to estimate ROC. GEE is an estimation method used in tandem with a specific model (linear, logistic). Was logistic regression used to build the ROC curves with GEE for longitudinal data analyses? Further, please specify which correlation structure was used with GEE (which should align with sampling methods and a priori hypothesis) as this can impact precision.

Answer: Thanks for pointing the need for more information about GEE. We used GEE with the logit link function. The correlation structure was compound symmetric (CS). We compared other correlation structures, but no changes were verified. Thus, we used CS attending the principle of parsimony. This information was included in the Statistical Analysis subsection.

“We compared drug levels between missing data for indirect measures and complete information using Generalized Estimating Equations (GEE) with the logit link function. The compound symmetry correlation structure was adopted for GEE”

16. I would add a citation for the Youdon index and a brief explanation of the purpose.

Answer: Thanks for this suggestion, we add a brief explanation of Youden Index and a citation. 

17. Throughout, sensitivity is sometimes referred to as sensibility and sometimes referred to as sensitivity. Please adjust all mentions of sensibility to sensitivity.

Answer: Thanks for pointing that. We correct it. 

18. You can not have over 100% for the pill count and self report as it is currently described. The “≥” [Line 150] should be changed to >.

Answer: Thanks for pointing that. We correct it. 

19. Please clarify what NA stands for in Table 1. Are these missing?

Answer: Thanks for pointing that. We indicate now that is information not available. This can be due to either missing data or a refusal to answer.

20. Please provide 95% confidence intervals for all estimates of AUC throughout the text considering the sample size is relatively small.

Answer: In Table 3, all 95% confidence intervals for AUC are provided. We avoid repeating this information in the text to enhance readability.

21. I would suggest supplemental tables that include values for the GEE analyses for Tables 4 and 5 to assess differences in the analytical methods. It seems as though overall in longitudinal analyses there are larger AUCs which may be reflective of behavior over time and if persons are being seen by the same provider, these metrics can be assessed.

Answer: The results in Tables 4 and 5 were not estimated using GEE; also we used cross-sectional data, as explained in the methods. We are presenting estimates of sensitivity, specificity, and predictive values using different cut-off points.

22. There exist instances in the text where aTGW and aMSM are used and others w

---

## [Decision Letter · Decision Letter 1]

9 Sep 2024

Performance of indirect adherence measures for daily oral pre-exposure prophylaxis for HIV among adolescent men who have sex with men and transgender women in Brazil.

PONE-D-24-10207R1

Dear Dr. Zeballos,

We’re pleased to inform you that your manuscript has been judged scientifically suitable for publication and will be formally accepted for publication once it meets all outstanding technical requirements.

Kind regards,

Fengyi Jin, Ph.D.

Academic Editor

PLOS ONE

Additional Editor Comments (optional):

Reviewers' comments:

Reviewer's Responses to Questions

**Comments to the Author**

1. If the authors have adequately addressed your comments raised in a previous round of review and you feel that this manuscript is now acceptable for publication, you may indicate that here to bypass the “Comments to the Author” section, enter your conflict of interest statement in the “Confidential to Editor” section, and submit your "Accept" recommendation.

Reviewer #1: All comments have been addressed

Reviewer #3: All comments have been addressed

2. Is the manuscript technically sound, and do the data support the conclusions?

Reviewer #1: Yes

Reviewer #3: Yes

3. Has the statistical analysis been performed appropriately and rigorously? 

Reviewer #1: Yes

Reviewer #3: Yes

4. Have the authors made all data underlying the findings in their manuscript fully available?

Reviewer #1: Yes

Reviewer #3: Yes

5. Is the manuscript presented in an intelligible fashion and written in standard English?

Reviewer #1: Yes

Reviewer #3: Yes

6. Review Comments to the Author

Reviewer #1: Thank you for the opportunity to review the revised manuscript. The authors did a great job addressing my feedback and I have no further comments.

Reviewer #3: The revised version looks good, and authors have responed all comments very well. I have no additional comments.

7. PLOS authors have the option to publish the peer review history of their article (what does this mean?). If published, this will include your full peer review and any attached files.

Reviewer #1: No

Reviewer #3: No

---

## [Editor Report · Acceptance letter]

18 Oct 2024

PONE-D-24-10207R1 

PLOS ONE

Dear Dr. Zeballos, 

I'm pleased to inform you that your manuscript has been deemed suitable for publication in PLOS ONE. Congratulations! Your manuscript is now being handed over to our production team.

Kind regards, 

on behalf of

Dr. Fengyi Jin 

Academic Editor

PLOS ONE